# *Hagnosa longicapillata*, gen. nov., sp. nov., a New Sordariaceous Ascomycete in the Indoor Environment, and the Proposal of Hagnosaceae fam. nov.

**DOI:** 10.3390/pathogens11050593

**Published:** 2022-05-18

**Authors:** Donát Magyar, András Tartally, Zsolt Merényi

**Affiliations:** 1National Public Health Center, 1097 Budapest, Hungary; 2Department of Evolutionary Zoology and Human Biology, University of Debrecen, 4032 Debrecen, Hungary; tartally.andras@science.unideb.hu; 3Synthetic and Systems Biology Unit, Institute of Biochemistry, Biological Research Center, 6726 Szeged, Hungary; zmerenyi@gmail.com

**Keywords:** *Hagnosa*, indoor fungi, spore dispersal

## Abstract

*Hagnosa longicapillata*, gen. nov., sp. nov, is described and illustrated from wooden building materials collected in Hungary and from pure culture. This species has been collected exclusively from indoor environments, where it was quite common. The ascocarps develop in a thick layer of brown, woolly mats of mycelia. The ostiolar region of the perithecia is ornamented with a five-lobed, flower-shaped crown. Asci are four-spored; ascospores are dark brown, smooth, muriform, not constricted at the septa, and liberated mostly through crackings of the thin ascomatal wall. Apparently, ascospores are dispersed by the mechanical disturbance of the mycelial web. In the phylogenetic tree, *Hagnosa* samples were placed as a basal lineage, independently from the other family of Sordariomycetidae, with high support. To place *Hagnosa* in Sordariales, the new family, Hagnosaceae, is proposed.

## 1. Introduction

The indoor environment provides optimal growth conditions for many fungi. The diversity of the fungal species on building materials is usually much lower than in the outdoors, due to the limited diversity of substrates and hosts. Wet building materials are often dominated by anamorphic genera, such as *Aspergillus* and *Penicillium*, in Europe [1]. Only a relatively few fungi are present as teleomorphs in buildings. Amongst them, the most common genera are *Aspergillus, Chaetomium, Microascus*, and *Ascotricha,* whereas *Cephalotheca, Myxotrichum, Ceratocystis, Sordaria*, and *Peziza* are less frequent (genera are listed in descending order of frequency, according to the authors’ observations in Hungary). Fungal contamination of indoor environments has recently become a major issue. Thus, public awareness of the potential negative effects of indoor fungi has increased in recent decades, especially among homeowners, facility managers, and the insurance industry [2]. As new species are relatively sparsely found in houses, discoveries attract the attention not only of mycologists, but also of hygienists and doctors. 

During studies of fungi inhabiting indoor environments, a fungal species with muriform, large spores, apparently belonging to a hitherto undescribed taxa, were found in Hungary. The fungus was isolated in pure culture, and is subsequently described here as a novel ascomycete genus and species. It is contrasted to members of other ascomycetous genera, based on multilocus phylogeny and morphological traits.

## 2. Results

### 2.1. Phylogenetic Analysis

To determine the genetic distance to other known species and to establish the phylogenetic position of our samples, BLAST searches and phylogenetic inferences were performed based on the nuclear ribosomal regions. Both the internal transcribed spacer (ITS) and the large subunit (LSU) were amplified and sequenced from spores and pure cultures. BLAST search with ITS and LSU sequences indicated that our isolates belong to Sordariomycetes. However, neither ITS (69–96% coverage and 80–85% identity) nor LSU (93–100% coverage and 91–94% identity) supported an exact species level match with the species deposited in the GenBank database (Appendix A). To place our isolates among Sordariomycetes, we extended the four-gene supermatrix of Hongsanan et al. (2017) [3] with LSU sequences of the samples. The nucleotide supermatrix comprised 12,746 aligned sites (including gaps) of the four loci (LSU, RPB2, SSU, TEF). As the best model, GTR+F+G4 was selected for LSU, SSU, and TEF, whereas GTR+F+I+G4 was selected for RBH2. In the phylogenetic tree (Figure 1), our specimens were placed as a basal lineage, independent of the other family of Sordariomycetidae with high support (80% ML). As a difference from the reference tree, it should be noted, that in our tree reconstruction, the Coniochaetaceae and Cordanaceae were not part of the Sordariomycetidae. These two clades were not even supported by the reference tree [3].

### 2.2. Taxonomy

Hagnosaceae D. Magyar and Z. Merényi fam. nov.

Saprobic on wood in terrestrial habitats. Sexual morph: Ascomata perithecial, dark brown to black, superficial, stroma absent, ovoid, globose to subglobose, tomentose. Ascospores muriform, brown, not constricted at the septa.

Asexual morph: unknown.

*Hagnosa* D. Magyar and Z. Merényi gen. nov.

*Etymology:* From ancient Greek, ἁγνός (hagnós, ‘chaste’). Ascomata immersed and hidden in wooly mycelia, ostiolate. The ostiolar region of the perithecia are ornamented with a five-lobed, flower-shaped crown. *Asci* four-spored. *Ascospores* muriform, brown, not constricted at the septa.

typus: Hagnosa longicapillata

*Hagnosa longicapillata* D. Magyar and Z. Merényi sp. nov. 

Figure 2 and Figure 3.

*Etymology*: from Latin *longicapillata* (‘long-haired’). 

Saprobic on wood. *Colonies* on natural substrates form thick (500–1700 µm wide) floccose, woolly, brown-greyish brown (RGB 77, 67, 62–127, 115, 103, Figure 2a) mats of mycelia with immersed ascocarps (Figure 2b). *Mycelia* are light brown to reddish brown, flexuous, septate, sparsely branched or knotted, smooth or slightly echinate, rarely anastomosing, sometimes in fascicles, 1.7–2.9 µm in diameter (on natural substrate, Figure 2f), 1.6–2.6 µm (on PDA). On artificial substrates: slowly growing, umbonate; on PDA, about 1.7–1.9 mm in diameter after 7 days and 15 mm in diameter after 8 months at 20 °C, with a floccose, woolly, brown, greyish brown (RGB 84, 73, 69–168, 153, 148) mycelium, with a thin dark brown rim having a slightly undulate edge, reverse dark brown, without exudates. Old cultures turn to reddish brown (RGB 76, 56, 55–81, 62, 58, Figure 2e). On MEA, colony morphology is similar to PDA (Figure 2d). On MN300, its growth is somewhat faster: 2 mm in diameter after 7 days. No growth on TAA, sawdust agar, and inoculated parquet tiles. On water, agar fungal growth was observed only when it was supplemented with soil, wood, or ant waste extract. On soil, MEA dark brown ascomata produced in 3 weeks under 8 h /day blacklight (310–410 nm UVA) illumination, but not on MEA, Soil-WA, and wood-extract agar at similar treatment. On the latter, ascocarps filled with oil droplets, but no spore production was observed. *Ascomata* perithecial, tomentose, immersed and hidden in densely matted woolly mycelia, pyriform, dark brown (Figure 2g–h and Figure 3a–d). On natural substrate, ascocarps are 207–250(–272) × (157–)200–230 µm; on Soil-MEA, (50–)60–95(–140) × (50–)70–110(–140) µm. *Corona:* ostioles are inconspicuous and surrounded by fascicles of short, 0–2-septate, curved, 15–16 × 2 µm hyphae, with unilaterally thickened and dark-pigmented wall (Figure 3e). Such hyphae are arranged to form five lobes, reminiscent of a flower shape (Figure 2i). The individual lobes are 18–21 µm long; the diameter and height of this corona is 54–89 µm and (23–)35–35(–41) µm, respectively; this head is connected to the ascocarp with a fragile, 47–66 µm wide neck. The corona was observed on natural substrate. *Ascomatal wall* thin, prosenchymatous, textura angularis, brown (Figure 2j). Ascocarps tend to detach from the substrate and are held by long mycelia, or develop on aerial mycelium (Figure 2b). Many of these long mycelia originate from the ascocarps. *Hamathecium* is composed of hymenial cells and hypal-like elements. Hymenial cells globose, pyriform, or ellipsoid 5.2–9.1 × 4.6–7.8 µm, in up to five layers, evanescent; cell walls of the lower layers undergo uneven pigmentation and thickening (Figure 2m), mostly at the distal parts of the wall and less at the middle. Such cells become rectangular (penta- and hexagonal, Figure 2k) and subsequently lose cytoplasm, forming a spongious layer (Figure 2n). On cell walls, pores surrounded by annular wall thickenings are observable (Figure 2l). Hypal-like elements (ascogenous hyphae?) evanescent, hyaline, septate, thin (0.8–2.2 µm), delicate, blurry, embedded in mucilage (Figure 2q). *Asci* evanescent, inamyloid, unitunicate, clavate with a short stalk and four irregularly-arranged, overlapping ascospores; ascospores completely fill each ascus, 75.5–78.2 × 28.9–29.8 µm (Figure 2o–p). *Ascospores* smooth, muriform, with (0–)5–6(–8) transverse and (0–)1–2(–3) longitudinal septa, not constricted at the septa; dark brown (RGB 87, 53, 50–120, 66, 49), broadly ellipsoid or clavate-pyriform, apical region sometimes protruding and bent, (22.4–)35–50(–56.9) × (9.8–)15–19 µm (on Soil-MEA, Figure 2u); on natural substrate: (28.5–)40–48(–52.4) × (17.4–)18–20(–28.3) µm (Figure 2s,t). Mature spores of markedly different sizes occur in the same ascocarp (2.5–17.5 vs. 28.3–49.1 µm). The spore wall is two-layered; the outer brown layer easily breaks by pressure, and separates from the inner, hyaline one (Figure 2u*). At lower magnification (~50×), ascospores appear to be shiny-black. 

*Holotype:* on old parquet hardwood; stored in a cellar, Szokolya; collected by D. Magyar on 29 August 2021; and deposited in the Hungarian Natural History Museum, Budapest (111779BP, barcode: HNHM-MYC 021541).

*Additional specimens examined:* on an old barrel, in a cellar, Szokolya, collected by D. Magyar on 12 August 2021 (111780BP, barcode: HNHM-MYC 021542, paratype); under old pine parquet, bedroom, Szokolya, collected by D. Magyar on 1 April 2021; on an old cupboard (made of hardwood (black locust-*Robinia pseudo*-*acacia*) stored in a cellar, Szigetújfalu, collected by P. Herke and M. Szabó on 19 April 2020 (111781BP, barcode: HNHM-MYC 021543, paratype); on wet, rotting parquet, Celldömölk, collected by D. Magyar on 30 June 2015; on old, rotting parquet, bedroom, Budapest, collected by D. Magyar on 8 September 2017; under old, rotting parquet, bedroom, Gyöngyös, collected by D. Magyar on 19 November 2019; on old wood, stored in a cellar, Pomáz, collected by D. Magyar on 28 September 2019; on wood boxes, in a champagne cellar, Budaörs, collected by Z. Tischner and D. Magyar on 9 January 2020; isolate on MEA, collected from a colony growing on wood boxes, in a champagne cellar, Budaörs, 9 January 2020, collected by D. Magyar and Z. Tischner, deposited in the Szeged Microbiological Collection (SZMC) at the Department of Microbiology, Faculty of Science and Informatics, University of Szeged, Hungary, SZMC 27682 (T726C).

### 2.3. Fungal Ecology and Dispersal

Common features of *Hagnosa’s* natural habitats were: low temperature (10–15 °C), very high relative humidity, no ventilation, and darkness. Substrate was dry (moisture content, 25%) old pine and hardwood, often attacked by house borers (*Anobium* sp.) and/or ants. The fungal colonies were found on processed wood (barrels, boxes, parquet tiles, and a cupboard) stored in cellars for several decades (Figure 4). However, it was also detected in extensively used rooms (living rooms, bedrooms, and in an elementary school). In a family house built in 1935, the fungus was found in both the cellar and the living room (under pine parquet) separated by a monolithic reinforced concrete floor with steel beams. It is thought that *Hagnosa* spores were transported by invading ants between the cellar and the living room. In the school building (built in 1927), the fungus was detected on the parquet in its gym hall. Although the parquet was dry, it become dark-brown and crumby, possibly because moist air rose from the flooded cellar below (with 10 cm water on the bottom). Ants (*Lasius* s.str.) and oribatid mites were found on this parquet too. In an old family house invaded by ants, the fungus was found under its old parquet, laid directly on the ground (soil) and covered by a new parquet. In a champagne cellar, oribatid mites were present in the colonies of *Hagnosa*.

The mean spore size of *Hagnosa* was compared with common indoor fungi, and shown in Figure 5. The length and width of *Hagnosa* spores (39.6 and 14.4 µm, respectively) were greater than those of other common indoor species. 

In the formicarium experiment, the fungus was intensively visited by ants (Figure 6, Appendix A). Large quantities of mycelial fragments of *Hagnosa* were found on visiting workers. Apparently, mycelium is difficult to remove from body parts (Appendix A). The ants were healthy, and their population number and activity did not decrease during the 25 days of the experiment. Workers harvested the fungal colony and formed 300–400 µm mycelial balls, containing some ascospores. Due to the agitation of the fungal colony by ants, ascospores were liberated, and a massive spore print was deposited around the fungus (width: 2–4 mm). Ant workers carried mycelial balls, held in their mandibles, to a distance of 10 cm, close the queen and the brood (’nest centre’; Appendix A) and inserted between the lid and the lower part of the Petri dish in 5 h. Textile fibers were also used by the ants as insulation material. Ant waste was deposited at the middle of the nest chamber as a ⌀ 5 mm flat mound, composed mostly of mycelia. The waste mound contained several (<95) ascospores, of which 10% were broken. Mycelia and ascospores were also found around the food and water supply, but scarcely. Attachment of mycelia and ascospores was also observed on *Formica cunicularia* (mostly on the legs and the gasteral hairs) when it touched the wooly mycelial layer (Appendix A).

## 3. Discussion

The most prominent feature of *Hagnosa* is the apparent morphological similarity of its ascospores to those of *Capronia.* Ascocarp morphology in *Hagnosa* is also somewhat similar to that in *Capronia*, as it is characterized by dark ascomata with ostioles often surrounded with a crown of short setae [4]. To some extent, the *H. longicapillata* is morphologically similar to *C. acutiseta* G. J. Samuels, having large, pigmented, ellipsoidal, muriform ascospores with 4–5 transverse and 1–2 longitudinal septa. However, it differs from *Hagnosa*, as it has acute setae, eight-spored, bitunicate, non-evanescent asci, paler (olivaceous) and shorter ascospores ((14–)17–25(–27) × (11–) 11.5–13 (–14) µm), constricted at the septa [5]. *Capronia rubiginosa*, another similar fungus, has a reddish brown oval to fusiform ascospores with obtuse ends, with 6–12 transverse septa, 1–10 longitudinal septa, and 0–3 oblique septa (18–35 × 7–15 µm). Molecular phylogenetic data show that *Hagnosa* differs to *Capronia* at a class level, as the latter genus belongs to Herpotrichiellaceae, Chaetothyriales, Eurotiomycetes. Dark-pigmented muriform ascospores are common in the Dothideomycetes class too, but these ascospores are characteristically constricted at the septa [6]. According to the molecular analysis, *Hagnosa* belongs to the Sordariomycetes. Muriform spores are rarely found in this class. Only one sordariaceous family, Thyridiaceae, contains such spores [7]. Unlike *Hagnosa*, Thyridiaceae are characterized by having stromata [7]. This family currently consists of two genera; the second genus, *Mattirolia* was recently added to this family by Checa et al. (2013) [8], even though stromata are not always present in some species. *Mattirolia mutabilis* is characterized by hyaline to yellow-greenish muriform ascospores, not constricted at the septa, as well as perithecia without a typical stroma; the perithecia are covered by a yellowish tomentum [8]. Although the morphologies of the perithecia and ascospores in *Mattirolia* are similar to that in *Hagnosa*, molecular analysis does not support its placement into *Thyridiaceae*. Based on the phylogenetic tree, the closest relative with similar morphologies of *Hagnosa* seems to be species from Chaetosphaeriaceae [9], but inclusion to this family can be rejected, as: (1) perithecia in this family are typically associated with conspicuous dematiaceous anamorphs, and (2) ascospores are always hyaline and have 1–3 transverse septa [10,11]. Therefore, to place *Hagnosa* in Sordariales, the new family, Hagnosaceae, is proposed. 

The fungus seems to be quite common in indoor environments, even in houses, where the fungus grows under old wet parquet. The fungal colonies are macroscopically similar to *Zasmidium cellare*. Cellars, especially those used to store wine, are typically covered with a thick mycelial layer of *Z. cellare* [12]. The macroscopic resemblance of *Hagnosa* to *Zasmidium* may be the reason why such a common species remained undiscovered in indoor environments. Ascocarps are hidden in the dense mycelial web. It is a quite unique feature of this fungus that most ascocarps are not connected to the substrate, but they are ‘held aloft’ by the mycelia. The mycelia growing from and around the ascocarps makes it similar to silkworm cocoons. Rarely, ascocarps are sitting on the substrate. It was frequently seen that spores were liberated after the breaking of the fragile thin wall of the ascocarp. Breaking takes place at different regions of the ascocarp: horizontally at the apical neck, but also at the central and basal region. Longitudinal breakings on the whole ascocarp body are also common. Most of the mature and young (one-celled, pale brown) spores are liberated dry, without the presence of mucilaginous contents or remnants of asci. Only in a few young ascocarps developed on artificial media was it possible to see some asci and hyphal-like elements in a mucilaginous matrix, but usually they disappeared before the breaking of the ascocarps and spore liberation. Free ascospores were scattered and deposited on the mycelial web around the broken ascocarp. 

Apparently, among common indoor fungi, *Hagnosa* has the largest spores (Figure 5) [13,14]. Most indoor fungi, such as *Aspergillus* and *Penicillium*, have small, one-celled, hyaline conidia, the so-called ‘penetrator’ type [15,16]. Their light, small spores remain afloat in the atmosphere for a long time (due to their low settling velocity), and are easily transported by gentle air currents inside buildings, between rooms and floors. *Hagnosa* spores contrast those of other indoor fungi, being large and heavy, unsuitable for airborne dispersal. To solve this paradox, a closer look is necessary on its morphological features. Once the ascospores are liberated, they are scattered nearby around the ascocarp. Ascospores are large enough to be trapped on the web of mycelia (Figure 2s,r). *Hagnosa* has long and flexible hyphae. Our initial (and unsuccessful) attempts to isolate spores from the mycelial web using a needle resulted in accidentally tensing hyphae, which catapulted spores and ascocarps to a distance of several centimeters. It seems that the dispersal strategy of this fungus’ spores is based on the mechanical disturbance of the mycelia. Insects passing on the mycelial web may pull the long, string-like hyphae (similarly to our needle). When one of these hyphae gets stuck to the moving insets’ leg, the hyphae get more and more bent and tense. Such tension stores potential energy. When the hyphae are released (or torn), the pulling force is quickly stopped, and the ascospore sitting on the hyphae is launched into the air. This process was demonstrated by pulling the legs of *Formica* on the *Hagnosa* colony. Remnants of insects (heads, legs, and antennae, mostly belonging to ants) were observed to be stuck in the mycelial web. Some other indoor fungi have long setae holding spore mass, such as *Ascotricha*, *Botryotrichum*, *Chaetomium*, and *Dichotomopilus* [17,18]. According to von Arx et al. (1986) [19], their spore mass is typically dispersed by beetles, ants, mites, and other animals. The mycelial web-holding ascospores surrounding *Hagnosa* ascocarps may have a similar function. One should think that these fungi may have an analogous dispersal strategy. These ‘long haired fungi’ are common in crawl spaces (under parquet, or behind drywall) where ventilation is limited, and thus, the dispersal of spores is not guaranteed by air currents. However, mechanical disturbance by ants or the resonance of footsteps on parquet, or the vibration of drywall by closing doors may provide enough kinetic energy to disperse spores. Our formicarium experiment demonstrated that workers of *L. niger* harvest and transport mycelia to the nest chamber to use it as an insulation material, like other fibers of non-fungal origin. In ants, it is a well-known behavior to regulate the microclimate of the nest by incorporating materials [20]. However, this behavior can be beneficial for *Hagnosa* as well. Its ascospores are dispersed by visiting ants to both short and long distances. Short distance dispersal of spores is achieved when ants are harvesting the mycelia. By cutting tense hyphae, ascospores are launched to the air in a high quantity, mostly to a short distance. Long distance transport (e.g., to 10 cm) of ascospores was also observed when ant workers carried mycelial balls. It is a question whether the fungus can be transported to further habitats by flying ant sexuals, e.g., by queens founding new colonies.

Hyphal fragments found in the food and water supply may be transported unintentionally as they attach to the hairs (mostly on the tarsus and the antennae) and tarsal claws. Ascospores transported long distances were also found, but to a lesser extent. From our current experiments, we cannot draw any conclusions about whether ants forage on *Hagnosa*. Most spores found in the nest’s waste mound were intact. In our larvae, brown faecal matter was observed, as is usual in ant larvae, but there was no evidence of a fungal origin. The attraction/avoidance response to fungi is a complex behavior [21]. Further experiments are therefore needed to see if larvae consume this fungus, and whether it has any effect on the reproduction rate of the ants. 

The obvious question about indoor fungi is whether they can cause illness in residents. In the case of *Hagnosa*, there is a small probability of breathing its large spores, thus they may have a negligible effect on respiratory health. 

The frequent findings of oribatid mites in *Hagnosa* colonies may not be a coincidence. Oribatid mites are known to preferentially feed on melanized fungi, and tend to reject hyaline forms [22], possibly because dark pigmented fungi contain more carbon or nutrients than other fungi [23]. It was speculated that oribatid mites preferentially disperse fungal species that they feed on, which would be an indication for a mutualistic relationship between oribatid mites and some fungal taxa [24]. Oribatid mite species preferentially feed on certain species of dark pigmented fungi, such as *Alternaria* [25]. Eggs possibly belonging to oribatid mites were also frequently found in the mycelial mat. Ants also seem to prefer melanized spores, such as *Alternaria* [26,27,28], *Bipolaris* and *Curvularia* [27,28], *Stemphylium*, *Helminthosporium* sensu lato [26], and *Oncopodiella* [28]. Interestingly, these fungi are similar to *Hagnosa* not only in dark pigmentation, but also in having large and multiseptate spores.

In *Hagnosa* colonies, a surprisingly high diversity of indoor fungal species was observed. Most frequently, *Bispora* sp., *Myxotrichum* sp., *Cephalotheca*-like sp., and an unknown ascomycete with 3-septate, striate ascospores were found there. Ascospores of *Aspergillus* and Polyporales were also common on the mycelial web. One can speculate that associated fungi may utilize the mycelial mat, offering more favorable microclimatic conditions for growth. Associated fungi may also take advantage of *Hagnosa* mycelia in dispersal, as their spores can also be launched by the flexuous hyphae. Further studies are needed to isolate and identify these species as well, in order to complete the knowledge of fungi living on old wooden materials in indoor environments. 

## 4. Materials and Methods

### 4.1. Isolation

Pieces of timber, covered by brown, woolly mycelia, were collected from water-damaged buildings and cellars. The moisture content (%) of wooden building materials (down to a depth of approx. 3 cm) were measured in situ with a Greisinger GMI 15 device (GHM Messtechnik GmbH, Remscheid, Germany). The samples were carried to the laboratory in sterile polyethylene bags. For the purpose of microscopic investigation, fungal structures were recovered from the substrate using a tape-lift method (with MACbond B 1200 pressure-sensitive acrylic strips, MACtac Europe S.A., Brussel, Belgium). The tape was pressed gently on the surface of 2% Malt Extract Agar (MEA). 1 × 1 cm blocks of MEA were cut out and placed onto a microscope slide. Ascospores occurring on the surface of MEA were detected with a Zeiss Axio Imager Z1 microscope under × 200 magnification, and gathered with a Wironit needle to culture on different artificial media and isolate DNA.

Digital photomicrographs were taken with a Zeiss Axio Imager Z1 microscope at × 400 magnification, and measured with CMEX5 Pro camera and Euromex Imagefocus alpha x64 software (Euromex Microscopen Bv, BD, Arnhem, the Netherlands). Fungal structures were mounted on glass slides with tap water for microscopic examination of living state [29] or methylene blue. To aid observations, mycelia were removed by rolling ascocarps on sellotape. 

Samples were inoculated to Malt Extract Agar medium (MEA; 30 g/L malt extract, 5 g/L peptone, 15 g/L agar) containing 0.1 g/L chloramphenicol, and incubated at 25 °C for 14–28 days. Growth tests were also performed on Potato Dextrose Agar (PDA; 39 g/L Potato dextrose agar (Difco), 0.01 g/L ZnSO_4_·7H_2_O, 0.005 g/L CuSO_4_·7H_2_O, pH 5.6); Water Agar medium (WA; 20 g/L agar); Tannic Acid Agar medium (TAA; 3.0 g/L NaNO_3,_ 1.0 g/L KH_2_PO_4_, 0.5 g/L MgSO_4_·7H_2_O, 0.5 g/L KCl, 0.01 g/L FeSO_4_·7H_2_O, 10 g/L Tannic acid; 30 g/L agar, pH 4.5); Soil Extract Agar medium with MEA or WA (Soil-MEA or Soil-WA; 10 g filter sterilized soil extract added to MEA or WA); Ant Waste Extract mediums (0.5 g filter sterilized ant waste extract added to WA; ant waste was collected from *Campinotus auriventris* and *Myrmecia pavida* formicariums); Wood Extract Agar medium (10 g filter sterilized wood extract added to MEA); Sawdust Agar (10 g fine pine parquet sawdust added to water agar); and MN300 media (0.8 g/L KNO_3_, 1 g/L K_2_HPO_4_, 0.5 g/L MgSO_4_.7H_2_O, 0.6 g/L NaCl; 0.5 g/L yeast extract, 0.5 g/L pepton (lactamin), 1 mL micronutrient solution; 10 g/L cellulose-pulver MN 300, 15 g/L agar). Artificial inoculation was also performed on sterilized parquet tiles in a moist chamber (a plastic bag with sterile distilled water). Colony colors were determined by the ‘RGB profiling’ procedure [30] of the ImageJ software ver. 1.34 s (RGB_Profiler.class plugin) [31] after 14 days at 25 °C on the bench. Reference strains are maintained in the Szeged Microbiological Collection (SZMC) at the University of Szeged, Hungary. The holotype is deposited in the Hungarian Natural History Museum, Budapest (BP).

### 4.2. Molecular Phylogeny

DNA was isolated from single spores from natural substrate, as well as from natural and pure cultures. The ascospores were transferred to 20 µL Milli-Q water in a PCR tube using a Wironit needle. DNA was amplified from single ascospores by direct PCR, following the procedure of O’Mahony et al. (2007) [32]. At first, the internal transcribed spacer (ITS) and the large subunit (LSU) were amplified using the primers ITS1F/LR5 [33,34]. After this PCR, we carried out further reactions to separate the ITS and LSU regions from each other with the primers ITS1F/ITS4 [33] and LR0R/LR5 [34], respectively. The PCR tubes were put into liquid nitrogen for around 10 s, and thereafter, into 95 °C heated PCR for around 20 s. This procedure was repeated 5–10 times. After this, there was a termination step at 95 °C for 3 min, and centrifugation. The PCR master-mix was added into these PCR tubes. PCRs were performed with a final volume of 50 μL; the components were as follows: DreamTaq Green Buffer (Fermentas) (20 mM MgCl_2_, 5.0 μL); dNTPmix (Fermentas) (2 mM, 5.0 μL); primers for ITS or LSU (0.01 mM, 1.0 μL); Milli-Q water (12.75 μL); DreamTaq polymerase (Fermentas) (5 unit/μL, 0.25 μL). Thermocycling was carried out under the following conditions: 94 °C for 5.5 min, 43 cycles of 94 °C for 18 s, annealing at 51 °C for 30 s, 72 °C for 45 s, and final extension at 72 °C for 7 min. Sanger sequencing was performed by Eurofins Genomics (GATC services, Konstanz, Germany). From the six trials, we obtained evaluable sequences from one ascospore (M2), and two samples which contained more ascospores and some mycelia (H1 and H2). We also sampled DNA from the mycelia of the strain collection using QIAGEN DNeasy Plant MiniKit for standard DNA extraction procedure.

Similar sequences were searched in GenBank (http://ncbi.nlm.nih.gov/, accessed on 25 May 2021) using the BLASTn search [35]. We obtained ITS and LSU sequences from our strains, and ITS sequences were used for the BLAST search, whereas LSU sequences were used for the phylogenetic analysis based on the combined LSU-SSU-TEF-RBP2 supermatrix. We used the four-loci supermatix from Hongsanan et al. [3] in order to classify our samples into a robust and reliable phylogenetic tree. The four loci were aligned with the L-INS-I algorithm of MAFFT [36] separately. We applied a trimming process to delete regions with more than 80% gaps, using TrimAl version 1.2 [37]. The suitable substitution matrix for phylogenetic analyses was selected with IQ-TREE [38]. Phylogenetic analysis under the maximum-likelihood (ML) criterion was also conducted with IQ-TREE 1.6.12 [38] using its fast bootstrap (1000×) option with subsequent search for the best tree, employing the selected models for each partition. The phylogenetic tree was visualized with FigTree version 1.3.1. [39]. Sequences obtained from *Hagnosa longicapillata* (herb no. T726C) were deposited with the accession numbers OL630146-OL630149 for LSU, and OL630154-OL630155 for ITS in GenBank.

### 4.3. Ecological Experiments

The mean spore size of *Hagnosa* was calculated and compared with common indoor fungi, of which data were obtained from the literature [13,14]. A scatter plot of length vs. width was created by R-package ggplot2 ‘3.3.5′ [40]. To test the interaction between the fungus and ants, an experimental formicarium was built in a plastic container (⌀ 180 mm, height 60 mm). Based on our observation (i.e., co-occurrence of *Hagnosa* and *Lasius niger* L. in Gyöngyös), *L. niger* was selected for the tests. A freshly overwintered young colony, containing one queen, 15 workers, and several larvae, was placed into the nest chamber of the formcarium. A Petri dish (⌀ 60 mm) was used as a nest chamber; its bottom was lined with dental gypsum (Fertisol, Budapest, Hungary) to hold moisture. Two openings, one at the lateral side (⌀ 2 mm) and one at the top (⌀ 5 mm), were cut on the Petri dish to allow ants to enter and to moisten the gypsum lining, respectively. To prevent housed ants from escaping the container, Fluon^®^ (AGC Chemicals America, Inc, Moorestown, NJ, USA), an aqueous polytetrafluoroethylene resin, was applied on the inner wall as a 40 mm width strip [41]. As food, 40% sucrose solution was provided in an Eppendorf tube; water was held in another container. A piece of the fungal colony on natural substrata (25 × 10 mm) was placed into the formicarium at a 20 mm distance from the nest chamber, the food, and the water supplies. As a control, textile fibers (80% wool, 20% nylon) of 5 mm length were also placed instead of the fungal colony, to compare the behavior of the ants against another fiber-like material of non-fungal origin. As *L. niger* workers are relatively small (3–5 mm), another ant species, *Formica cunicularia* Latreille (4–6.5 mm), was used to test the attachment of *Hagnosa* hyphae onto the ant’s body. The insect, held in forceps, was touched 10 times to the surface of the fungal colony (on natural substrate), and then examined under a light microscope at 40–400× magnification. 

## Figures and Tables

**Figure 1 pathogens-11-00593-f001:**
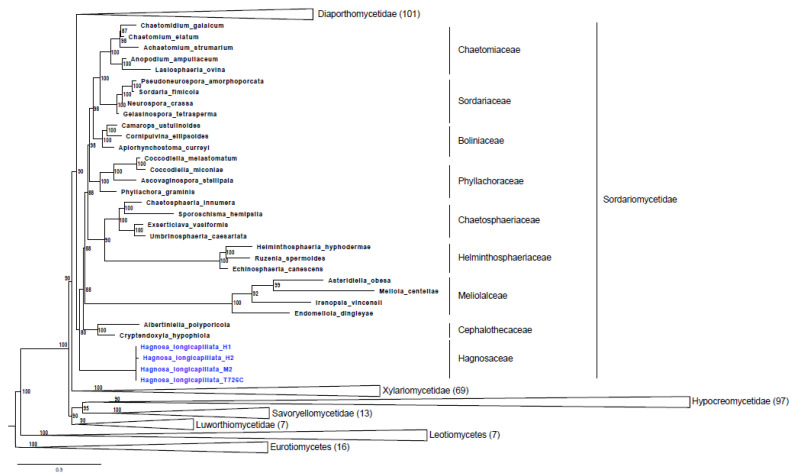
Maximum likelihood phylogenetic tree of Sordariomycetes adapted from Hongsanan et al. (2017) [3]. Distant lineages from *Hagnosa longicapillata* were collapsed, and the numbers in brackets represent the number of collapsed specimens. The scale bar means 0.3 expected nucleotide changes in both per site and per branch. Only values where ML bootstrap support >70% are shown. DNA extracted from ascospores and mycelia isolated from natural substrate (H1 and H2), from single ascospores isolated from natural substrate (M2), and mycelia from artificial culture (T726C).

**Figure 2 pathogens-11-00593-f002:**
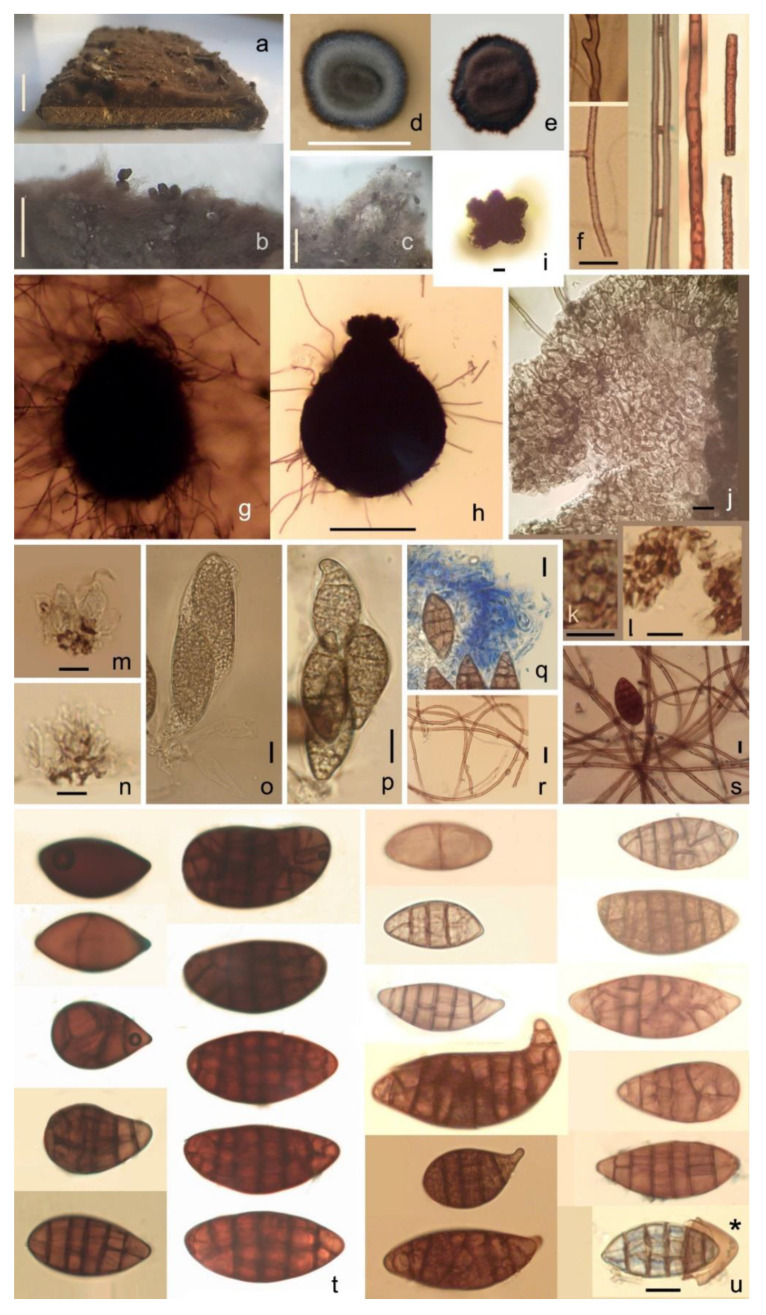
*Hagnosa longicapillata*. (**a**–**c**): colony on natural substrate; (**d**): colony on MEA; (**e**): old colony on Soil-MEA; (**f**): hyphae; (**g**): ascocarp; (**h**): ascocarp (hair removed); (**i**): corona; (**j**): ascomatal wall; (**k**): hexagonal cells; (**l**): pores with annular thickening; (**m**): hymenial cells; (**n**): hymenial cells after losing cytoplasm; (**o**): young ascus; (**p**): mature ascus; (**q**): hyphal-like elements in a mucilaginous matrix; (**r**): tensed, flexible mycelia; (**s**): ascospore on mycelia, ready to launch; (**t**): ascospores on natural substrate; (**u**): ascospores on Soil-MEA. Asterisk: broken ascospore wall, separating outer brown layer. Scale bars: (**a**): 10 mm, (**b**,**c**): 1 mm, (**f**): 10 µm, (**g**,**h**): 100 µm, (**i**–**u**): 10 µm. Stained with methylene blue, except (**o**,**p**), which are mounted in tap water.

**Figure 3 pathogens-11-00593-f003:**
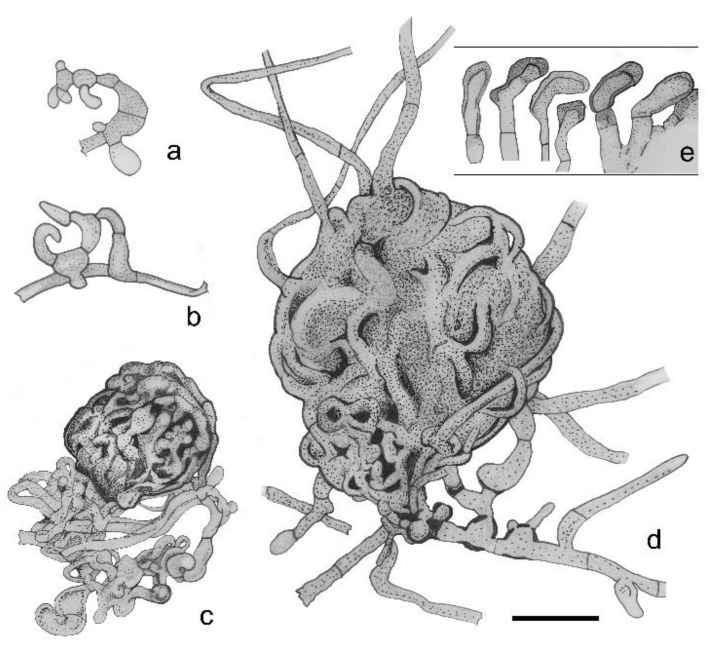
*Hagnosa longicapillata*. (**a**–**d**): Ascomata development. (**a**,**b**): ascogonium, (**c**,**d**): protoperithecium; (**e**): hyphae of the corona. Scale bar = 10 µm.

**Figure 4 pathogens-11-00593-f004:**
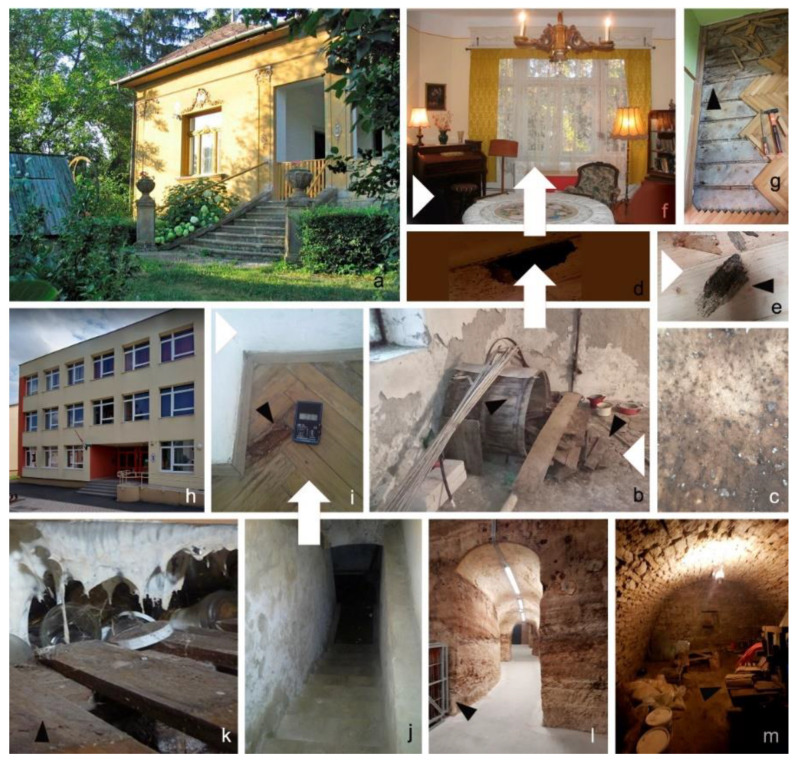
Habitats of *Hagnosa longicapillata*. (**a**–**f**): the author’s house; (**b**): fungal colonies on a barrel and parquet tiles in the cellar; (**c**): young colony on the barrel; (**d**): hole on the old, broken parquet; and (**e**): fungal colony recovered from it. The position of the hole on the floor of the (**f**): bedroom. (**g**): Fungal colony in another family house on old parquet, covered by a new layer of parquet. (**h**–**j**): Elementary school. (**i**): Colonies on parquet tiles of school gym hall with a moisture meter. (**j**): Flooded cellar under the gym hall. (**k**): Colonies on a cupboard, in a cellar. (**l**): Colonies on wooden boxes in a champagne cellar. (**m**): Colonies on old wood in a cellar. White arrows show the possible transfer of humidity; black arrows show fungal colonies.

**Figure 5 pathogens-11-00593-f005:**
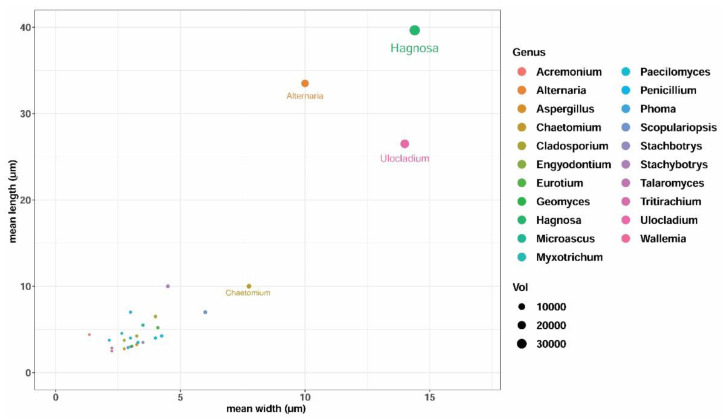
Mean length and width of spores of fungi common on indoor building materials. Each dot represents a species, colored by their genus. The size of the dots correlate with the volume of spores calculated with the formula of the spheroid. Displayed species: *Acremonium strictum*, *Alternaria tenuissima*, *Aspergillus glaucus* (teleomorph), *Asp. nidulans*, *Asp. niger*, *Asp. ochraceus*, *Asp. sydowii*, *Asp. versicolor*, *Chaetomium globosum*, *Cladosporium cladosporioides*, *Cl. herbarum*, *Cl. sphaerospermum*, *Engyodontium album*, *Geomyces pannorum*, *Hagnosa longicapillata*, *Microascus cirrosus*, *M. brevicaulis* (anamorph) *Myxotrichum defelxum*, *Paecilomyces variotii*, *Penicillium brevicompactum*, *P. chrysogenum*, *P. commune*, *P. palitans*, *P. variabile*, *Phoma glomerata*, *Stachbotrys echinata*, *S. chartarum*, *Talaromyces macrosporus*, *Tritirachium oryzae*, *Ulocladium alternariae*, *Wallemia sebi*.

**Figure 6 pathogens-11-00593-f006:**
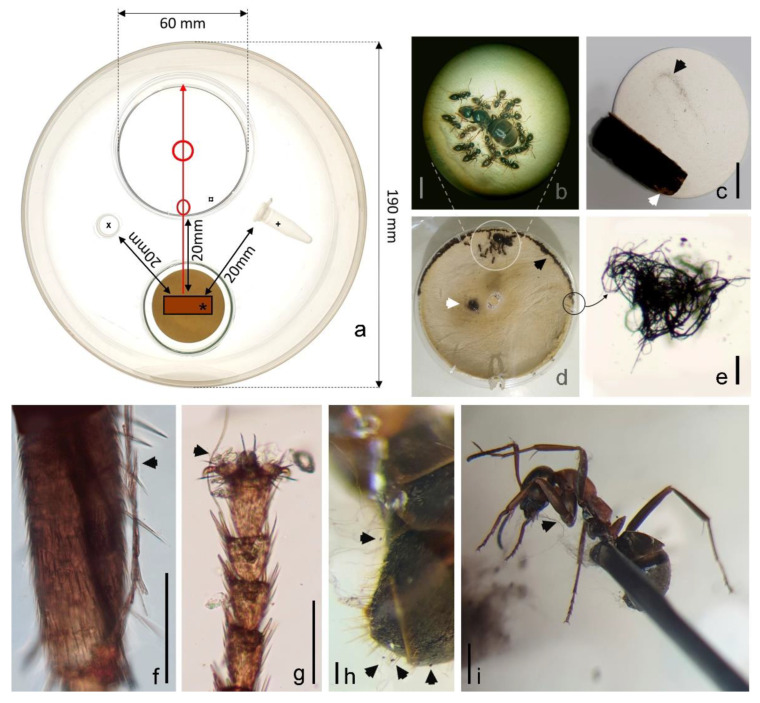
Formicarium experiments. (**a**): experimental settings of the formicarium in a plastic container. *: fungal colony, x: water, +: food (40% sucrose solution), ¤: nest chamber in a Petri dish, red circles: openings on the nest chamber, red arrow: route of transport of the fungal material by the ants. (**b**): ant colony, the workers, and the queen. (**c**): fungal colony used in the experiment on natural substrate (wood); white arrowhead shows the surface harvested by the ants, and the black arrowhead shows spore print around the original place of the fungal colony. (**d**): nest chamber with the ant colony; white arrowhead: waste mound; black arrowhead: insulation made of mycelial balls. (**e**): mycelial ball used to insulate the nest chamber. (**f**,**g**): tarsal hairs and claw of *Lasius niger*; black arrowheads show attached hyphal fragments. (**h**,**i**): *Formica cunicularia* after touching *Hagnosa* colony. (**h**): gasteral hairs; black arrowheads: ascospores. (**i**): mycelia on various body parts; black arrowhead shows attached mycelia. Scale bars: (**b**): 2 mm, (**c**): 10 mm, (**e**): 100 µm, (**f**): 50 µm, (**g**,**h**): 100 µm, (**i**): 1 mm.

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
