# Peer review of "Hagnosa longicapillata, gen. nov., sp. nov., a New Sordariaceous Ascomycete in the Indoor Environment, and the Proposal of Hagnosaceae fam. nov."

_pathogens, 2022, doi:10.3390/pathogens11050593_

Round 1
Reviewer 1 Report
I have carefully read MS which was submitted for consideration in the Pathogens (MDPI). In the article, the authors described a newly discovered genus Hagnosa and new species H. longicapillata of indoor fungus, belonging to the Ascomycota. In my opinion, this article in its current version is not suitable for publication. Besides, I believe that Pathogens is not an appropriate journal to publish information about the newly described taxa, I recommend choosing a different, less prestigious journal.
The article needs significant work on expanding the background to help new students or peers in related fields to understand aspects of the authors' discovery and description of new taxa. In my opinion, the Discussion chapter is poorly written and does not contribute much to the development of science, it should be improved. Moreover, this manuscript needs some revision, especially for language.
Specific comments:
Please include the dedication (for the wife) in the Acknowledgments section.
The title should be changed, it is unprofessional.
Lines 26-27: “Building materials are dominated my anamorphic genera, such as Aspergillus and Penicillium [1]” - Sentence requires rephrasing
Please add a short introduction to the results (at the end of chapter 1.) It is not clear what the presented results are about, why were genetic analyzes performed, what were they supposed to explain?
Lines 159-161: “Authors should discuss the results and how they can be interpreted from the perspective of previous studies and of the working hypotheses. The findings and their implications should be discussed in the broadest context possible. Future research directions may also be highlighted.” - I am confused by this sentence! Please remove this fragment, it's a technical manual!
Line 255: Figure 5. In my opinion, this figure has no scientific value, of course, it can be published, but rather in a popular science journal.
Author Response
Review 1
Comments and Suggestions for Authors
I have carefully read MS which was submitted for consideration in the Pathogens (MDPI). In the article, the authors described a newly discovered genus Hagnosa and new species H. longicapillata of indoor fungus, belonging to the Ascomycota. In my opinion, this article in its current version is not suitable for publication. Besides, I believe that Pathogens is not an appropriate journal to publish information about the newly described taxa, I recommend choosing a different, less prestigious journal.
Response: Thank you for you for reviewing our manuscript. We have sumbitted it to MDPI Pathogens, Special Issue: Detection of Indoor Fungi, because it fits tits scope. Herewith we detected and described a new indoor fungus, belonging to a new genera and family. Prestigious mycological journals usually rejec the publication of single species, but this is not the case. Herewith we are describing a new genus and family, and observations on the indoor dispersal of the spores. We hope that the Editors of MDPI Pathogens will accept our study for publication.
The article needs significant work on expanding the background to help new students or peers in related fields to understand aspects of the authors' discovery and description of new taxa. In my opinion, the Discussion chapter is poorly written and does not contribute much to the development of science, it should be improved. Moreover, this manuscript needs some revision, especially for language.
Response: Thank you. We reconsidered the text and made some corrections, marked with track changes.
Specific comments:
Please include the dedication (for the wife) in the Acknowledgments section.
Response: Thank you for this observation. According to the Instruction for Authors the section „Acknowledgments” contains „any support given which is not covered by the author contribution or funding sections. This may include administrative and technical support, or donations in kind (e.g., materials used for experiments).” https://www.mdpi.com/journal/pathogens/instructions. As you can see, dedications should not be included here. Usually, dedications are on the first page in MDPI articles, e.g. https://mdpi-res.com/d_attachment/ijms/ijms-17-00680/article_deploy/ijms-17-00680.pdf. We will ask the Editors’s opinion.
The title should be changed, it is unprofessional.
Response: Thank you. We accept your observation and changed the title according to similar papers, e.g. https://academic.oup.com/femsle/article/247/2/161/490993?login=true and https://www.microbiologyresearch.org/content/journal/ijsem/10.1099/ijs.0.056812-0 .
The new title is:
Hagnosa longicapillata, gen. nov. and, sp. nov., a new sordariaceous ascomycete in indoor environment, and the proposal of in Hagnosaceae fam. nov., found in indoor environment
Lines 26-27: “Building materials are dominated my anamorphic genera, such as Aspergillus and Penicillium [1]” - Sentence requires rephrasing
Response: Thank you, the sentence was rewritten: ’Wet building materials are often dominated my anamorphic genera, such as Aspergillus and Penicillium in Europe’
Please add a short introduction to the results (at the end of chapter 1.) It is not clear what the presented results are about, why were genetic analyzes performed, what were they supposed to explain?
Response: Thank you for your suggestion, we added some sentences as an introduction for molecular analysis. ’To infer/estimate for the genetic distance from other known species and to find the phylogenetic position of our samples BLAST searches and phylogenetic inference were carried out based on nuclear ribosomal regions. Both the internal transcribed spacer (ITS) and the large subunit (LSU) were extracted, amplified and sequenced from conidia and pure culture as well.’ This introduction is placed into the Materials and Methods part, because it better fints here than in the Introduction chapter.
Lines 159-161: “Authors should discuss the results and how they can be interpreted from the perspective of previous studies and of the working hypotheses. The findings and their implications should be discussed in the broadest context possible. Future research directions may also be highlighted.” - I am confused by this sentence! Please remove this fragment, it's a technical manual!
Response: Thank you for bringing this error to our attention. The text has been left from the template. It has been deleted.
Line 255: Figure 5. In my opinion, this figure has no scientific value, of course, it can be published, but rather in a popular science journal.
Response: Thank you for sharing your opinion with us. Artistic representation is widely used in science, most commonly in planetology (e.g. doi: 10.1126/science.abj9510), or in medical sciences (https://doi.org/10.1016/j.mex.2019.03.016). In mycology, such artworks are published since the 1960’s in journals like Mycotaxon. A prominent example is „Nematode-exploiting fungi - a microscopic landscape, with Arthrobotrys oligospora” see http://www.mycolog.com/chapter15.html. Another exaple is „The dispersal of mold spores by feeding dust mites” see https://link.springer.com/chapter/10.1007/978-3-319-29137-6_14. According to Khoury et al. (2019), „graphics have the potential to increase the attractiveness, understandability, and communication power of research findings.” Khoury, C. K., Kisel, Y., Kantar, M., Barber, E., Ricciardi, V., Klirs, C., ... & Novy, A. (2019). Science–graphic art partnerships to increase research impact. Communications Biology, 2(1), 1-5. However, as this topic is rather subjective, we leave the decision to the Publisher.
Reviewer 2 Report
Manuscript ID: pathogens-1619059
Title: Hagnosa longicapillata, gen. nov. and sp. nov., a new sordariaceous ascomycete in Hagnosaceae fam. nov., found in indoor environment.
It is a nice little paper describing an interesting new fungus from indoor environments. The only thing this reviewer would have wished for was growth descriptions (if any growth) on V8 and DG18, but only because this reviewer use the media regularly.
The paper reads well and where are only few typing errors and other thing to correct. Specific issues to be dealt with:
L28: Check genus names in e.g. indexfungorum for current names. Eurotium is now Aspergillus
L159-162: Delete the paragraph
L280: such as Alternaria….
L284: Helminthosporium
L319-320: Delete sentence
Author Response
Review 2
Title: Hagnosa longicapillata, gen. nov. and sp. nov., a new sordariaceous ascomycete in Hagnosaceae fam. nov., found in indoor environment.
Response: Thank you for you for reviewing our manuscript
It is a nice little paper describing an interesting new fungus from indoor environments. The only thing this reviewer would have wished for was growth descriptions (if any growth) on V8 and DG18, but only because this reviewer use the media regularly.
Response: Thank you for this excellent idea. We plated the fungus onto DG18. However, the fungus is growing slowly, and our 10 days limit to preparing responses does not allow us to compare the morphology with well-developed cultures on PDA and other media. Especially the development of ascomata would be informative, but it takes 3 weeks on other culture media. We are continuing tests with this fungus, and the results on DG18 are intended to be published in the future.
The paper reads well and where are only few typing errors and other thing to correct. Specific issues to be dealt with:
L28: Check genus names in e.g. indexfungorum for current names. Eurotium is now Aspergillus
Response: Thank you, corrected.
L159-162: Delete the paragraph
Response: Thank you, deleted.
L280: such as Alternaria….
Response: Thank you, corrected.
L284: Helminthosporium
Response: Thank you, corrected.
L319-320: Delete sentence
Response: Thank you, deleted.
Reviewer 3 Report
Dear Authors,
in my opinion your work is very interesting in a cognitive context and contributes a lot to mycology and evolutionary taxonomy. Authors report on the isolation and full characterization for Hagnosa longicapillata, the new species within the family of Sordariomycetidae.
All the figures are appropriate for this type of article. In general, the paper has a logical flow and fit the aims and scope of the journal. The abstract well correspond with the main aspects of the work. I do not see any significant shortcomings in this work and moreover I believe that it is refined in every detail.
As a reviewer I am also obligated to pay attention to even the smallest details. For this reason, I only request three cosmetic corrections.
Line 129
There is ,,ascosopres” but should be ,,ascospores”
Line 153, Figure 3
The mycelial structures designated a, b, c and e have been described. What about d ?
Line 358
MgCl2 should be written with subscript
Author Response
Review 3
Dear Authors,
in my opinion your work is very interesting in a cognitive context and contributes a lot to mycology and evolutionary taxonomy. Authors report on the isolation and full characterization for Hagnosa longicapillata, the new species within the family of Sordariomycetidae.
All the figures are appropriate for this type of article. In general, the paper has a logical flow and fit the aims and scope of the journal. The abstract well correspond with the main aspects of the work. I do not see any significant shortcomings in this work and moreover I believe that it is refined in every detail.
As a reviewer I am also obligated to pay attention to even the smallest details. For this reason, I only request three cosmetic corrections.
Response: Thank you for you for reviewing our manuscript
Line 129 There is ,,ascosopres” but should be ,,ascospores”
Response: Thank you, corrected.
Line 153, Figure 3 The mycelial structures designated a, b, c and e have been described. What about d ?
Response: Thank you for this observation. d is a protoperithecium. We corrected the text.
Line 358 MgCl2 should be written with subscript
Response: Thank you, corrected.
Round 2
Reviewer 1 Report
The authors re-submitted their paper on Hagnosa longicapillata, gen. nov., sp. nov. The paper has been tidied up, but unfortunately, the main issue of data quality and questionable interpretation remains. This is still a taxonomic article that should be published in another journal. Overall, in my opinion, the current data set is not appropriate for publication in Pathogens, and the conclusions do not have support in the data presented. Unfortunately, I must, again, strongly recommend not to accept this paper for publication, nor to be re-submitted with the current dataset.
Author Response
Thank you for your review. The manuscript was re-written according to the Editor's recommendatons. : A new test (formicarium experiment) was performed. We added its results, method and discussion to the manuscript. Former speculations were deleted. The results shows both short and long distance dispersal of ascospores by the ants.